# SOM-VAE: Interpretable Discrete Representation Learning on Time Series

**Vincent Fortuin, Matthias Hüser & Francesco Locatello**
Department of Computer Science, ETH Zürich
Universitätsstrasse 6, 8092 Zürich, Switzerland
`{fortuin, mhueser, locatelf}@inf.ethz.ch`

**Heiko Strathmann**
Gatsby Unit, University College London
25 Howland Street, London W1T 4JG, United Kingdom
`heiko.strathmann@gmail.com`

**Gunnar Rätsch**
Department of Computer Science, ETH Zürich
Universitätsstrasse 6, 8092 Zürich, Switzerland
`raetsch@inf.ethz.ch`

## Abstract

High-dimensional time series are common in many domains. Since human cognition is not optimized to work well in high-dimensional spaces, these areas could benefit from interpretable low-dimensional representations. However, most representation learning algorithms for time series data are difficult to interpret. This is due to non-intuitive mappings from data features to salient properties of the representation and non-smoothness over time.

To address this problem, we propose a new representation learning framework building on ideas from interpretable discrete dimensionality reduction and deep generative modeling. This framework allows us to learn discrete representations of time series, which give rise to smooth and interpretable embeddings with superior clustering performance. We introduce a new way to overcome the non-differentiability in discrete representation learning and present a gradient-based version of the traditional self-organizing map algorithm that is more performant than the original. Furthermore, to allow for a probabilistic interpretation of our method, we integrate a Markov model in the representation space. This model uncovers the temporal transition structure, improves clustering performance even further and provides additional explanatory insights as well as a natural representation of uncertainty.

We evaluate our model in terms of clustering performance and interpretability on static (Fashion-)MNIST data, a time series of linearly interpolated (Fashion-)MNIST images, a chaotic Lorenz attractor system with two macro states, as well as on a challenging real world medical time series application on the eICU data set. Our learned representations compare favorably with competitor methods and facilitate downstream tasks on the real world data.

## 1 Introduction

Interpretable representation learning on time series is a seminal problem for uncovering the latent structure in complex systems, such as chaotic dynamical systems or medical time series. In areas where humans have to make decisions based on large amounts of data, interpretability is fundamental to ease the human task. Especially when decisions have to be made in a timely manner and rely on observing some chaotic external process over time, such as in finance or medicine, the need for intuitive interpretations is even stronger. However, many unsupervised methods, such as clustering, make misleading *i.i.d.* assumptions about the data, neglecting their rich temporal structure and smooth behaviour over time. This poses the need for a method of clustering, where the clusters assume a topological structure in a lower dimensional space, such that the representations of the time series retain their smoothness in that space. In this work, we present a method with these properties.

We choose to employ deep neural networks, because they have a very successful tradition in representation learning (Bengio et al., 2013). In recent years, they have increasingly been combined with generative modeling through the advent of generative adversarial networks (GANs) (Goodfellow et al., 2014) and variational autoencoders (VAEs) (Kingma and Welling, 2013). However, the representations learned by these models are often considered cryptic and do not offer the necessary interpretability (Chen et al., 2016). A lot of work has been done to improve them in this regard, in GANs (Chen et al., 2016) as well as VAEs (Higgins et al., 2017; Esmaeili et al., 2018). Alas, these works have focused entirely on continuous representations, while discrete ones are still underexplored.

In order to define temporal smoothness in a discrete representation space, the space has to be equipped with a topological neighborhood relationship. One type of representation space with such a structure is induced by the self-organizing map (SOM) (Kohonen, 1990). The SOM allows to map states from an uninterpretable continuous space to a lower-dimensional space with a predefined topologically interpretable structure, such as an easily visualizable two-dimensional grid. However, while yielding promising results in visualizing static state spaces, such as static patient states (Tirunagari et al., 2015), the classical SOM formulation does not offer a notion of time. The time component can be incorporated using a probabilistic transition model, e.g. a Markov model, such that the representations of a single time point are enriched with information from the adjacent time points in the series. It is therefore potentially fruitful to apply the approaches of probabilistic modeling alongside representation learning and discrete dimensionality reduction in an end-to-end model.

In this work, we propose a novel deep architecture that learns topologically interpretable discrete representations in a probabilistic fashion. Moreover, we introduce a new method to overcome the non-differentiability in discrete representation learning architectures and develop a gradient-based version of the classical self-organizing map algorithm with improved performance. We present extensive empirical evidence for the model's performance on synthetic and real world time series from benchmark data sets, a synthetic dynamical system with chaotic behavior and real world medical data.

Our main contributions are to

- Devise a novel framework for interpretable discrete representation learning on time series.
- Show that the latent probabilistic model in the representation learning architecture improves clustering and interpretability of the representations on time series.
- Show superior clustering performance of the model on benchmark data and a real world medical data set, on which it also facilitates downstream tasks.

## 2 PROBABILISTIC SOM-VAE

Our proposed model combines ideas from self-organizing maps (Kohonen, 1990), variational autoencoders (Kingma and Welling, 2013) and probabilistic models. In the following, we will lay out the different components of the model and their interactions.

### 2.1 INTRODUCING TOPOLOGICAL STRUCTURE IN THE LATENT SPACE

A schematic overview of our proposed model is depicted in Figure 1. An input $x \in \mathbb{R}^d$ is mapped to a latent encoding $z_e \in \mathbb{R}^m$ (usually $m < d$) by computing $z_e = f_\theta(x)$, where $f_\theta(\cdot)$ is parameterized by the encoder neural network. The encoding is then assigned to an embedding $z_q \in \mathbb{R}^m$ in the dictionary of embeddings $E = \{e_1, \ldots, e_k \mid e_i \in \mathbb{R}^m\}$ by sampling $z_q \sim p(z_q|z_e)$. The form of this distribution is flexible and can be a design choice. In order for the model to behave similarly to the original SOM algorithm (see below), in our experiments we choose the distribution to be categorical with probability mass 1 on the closest embedding to $z_e$, i.e. $p(z_q|z_e) = \mathbb{1}[z_q = \arg\min_{e \in E} \|z_e - e\|^2]$, where $\mathbb{1}[\cdot]$ is the indicator function. A reconstruction $\hat{x}$ of the input can then be computed as $\hat{x} = g_\phi(z)$, where $g_\phi(\cdot)$ is parameterized by the decoder neural network. Since the encodings and embeddings live in the same space, one can compute two different reconstructions, namely $\hat{x}_e = g_\phi(z_e)$ and $\hat{x}_q = g_\phi(z_q)$.

To achieve a topologically interpretable neighborhood structure, the embeddings are connected to form a self-organizing map. A self-organizing map consists of $k$ nodes $V = \{v_1, \ldots, v_k\}$, where every node corresponds to an embedding in the data space $e_v \in \mathbb{R}^d$ and a representation in a lower-dimensional discrete space $m_v \in M$, where usually $M \subset \mathbb{N}^2$. During training on a data set $\mathcal{D} = \{x_1, \ldots, x_n\}$, a winner node $\tilde{v}$ is chosen for every point $x_i$ according to $\tilde{v} = \arg\min_{v \in V} \|e_v - x_i\|^2$. The embedding vector for every

Figure 1: Schematic overview of our model architecture. Time series from the data space [green] are encoded by a neural network [black] time-point-wise into the latent space. The latent data manifold is approximated with a self-organizing map (SOM) [red]. In order to achieve a discrete representation, every latent data point $(z_e)$ is mapped to its closest node in the SOM $(z_q)$. A Markov transition model [blue] is learned to predict the next discrete representation $(z_q^{t+1})$ given the current one $(z_q^t)$. The discrete representations can then be decoded by another neural network back into the original data space.

node $u \in V$ is then updated according to $e_u \leftarrow e_u + N(m_u, m_{\tilde{v}}) \eta (x_i - e_u)$, where $\eta$ is the learning rate and $N(m_u, m_{\tilde{v}})$ is a neighborhood function between the nodes defined on the representation space $M$. There can be different design choices for $N(m_u, m_{\tilde{v}})$. A more thorough review of the self-organizing map algorithm is deferred to the appendix (Sec. A).

We choose to use a two-dimensional SOM because it facilitates visualization similar to Tirunagari et al. (2015). Since we want the architecture to be trainable end-to-end, we cannot use the standard SOM training algorithm described above. Instead, we devise a loss function term whose gradient corresponds to a weighted version of the original SOM update rule (see below). We implement it in such a way that any time an embedding $e_{i,j}$ at position $(i, j)$ in the map gets updated, it also updates all the embeddings in its immediate neighborhood $N(e_{i,j})$. The neighborhood is defined as $N(e_{i,j}) = \{e_{i-1,j}, e_{i+1,j}, e_{i,j-1}, e_{i,j+1}\}$ for a two-dimensional map.

The loss function for a single input $x$ looks like

$$\mathcal{L}_{\text{SOM-VAE}}(x, \hat{x}_q, \hat{x}_e) = \mathcal{L}_{\text{reconstruction}}(x, \hat{x}_q, \hat{x}_e) + \alpha \, \mathcal{L}_{\text{commitment}}(x) + \beta \, \mathcal{L}_{\text{SOM}}(x) \qquad (1)$$

where $x$, $z_e$, $z_q$, $\hat{x}_e$ and $\hat{x}_q$ are defined as above and $\alpha$ and $\beta$ are weighting hyperparameters.

Every term in this function is specifically designed to optimize a different model component. The first term is the reconstruction loss $\mathcal{L}_{\text{reconstruction}}(x, \hat{x}_q, \hat{x}_e) = \|x - \hat{x}_q\|^2 + \|x - \hat{x}_e\|^2$. The first subterm of this is the discrete reconstruction loss, which encourages the assigned SOM node $z_q(x)$ to be an informative representation of the input. The second subterm encourages the encoding $z_e(x)$ to also be an informative representation. This ensures that all parts of the model have a fully differentiable credit assignment path to the loss function, which facilitates training. Note that the reconstruction loss corresponds to the evidence lower bound (ELBO) of the VAE part of our model (Kingma and Welling, 2013). Since we assume a uniform prior over $z_q$, the KL-term in the ELBO is constant w.r.t. the parameters and can be ignored during optimization.

The term $\mathcal{L}_{\text{commitment}}$ encourages the encodings and assigned SOM nodes to be close to each other and is defined as $\mathcal{L}_{\text{commitment}}(x) = \|z_e(x) - z_q(x)\|^2$. Closeness of encodings and embeddings should be expected to already follow from the $\mathcal{L}_{\text{reconstruction}}$ term in a fully differentiable architecture. However, due to the non-differentiability of the embedding assignment in our model, the $\mathcal{L}_{\text{commitment}}$ term has to be explicitly added to the objective in order for the encoder to get gradient information about $z_q$.

The SOM loss $\mathcal{L}_{\text{SOM}}$ is defined as $\mathcal{L}_{\text{SOM}}(x) = \sum_{\tilde{e} \in N(z_q(x))} \|\tilde{e} - \text{sg}[z_e(x)]\|^2$, where $N(\cdot)$ is the set of neighbors in the discrete space as defined above and $\text{sg}[\cdot]$ is the gradient stopping operator that does not change the outputs during the forward pass, but sets the gradients to $0$ during the backward pass. It encourages the neighbors of the assigned SOM node $z_q$ to also be close to $z_e$, thus enabling the embeddings to exhibit a self-organizing map property, while stopping the gradients on $z_e$ such that the encoding is not pulled in the direction of the neighbors. This term enforces a neighborhood relation between the discrete codes and encourages all SOM nodes to ultimately receive gradient information from the data. The gradient stopping in this term is motivated by the observation that the data points themselves do not get moved in the direction of their assigned SOM node's neighbors in the original SOM algorithm either (see above). We want to optimize the embeddings based on their neighbors, but not the respective encodings, since any single encoding should be as close as possible to its assigned embedding and not receive gradient information from any other embeddings

that it is not assigned to. Note that the gradient update of a specific SOM node in this formulation depends on its distance to the encoding, while the step size in the original SOM algorithm is constant. It will be seen that this offers some benefits in terms of optimization and convergence (see Sec. 4.1).

## 2.2 Overcoming the Non-Differentiability

The main challenge in optimizing our architecture is the non-differentiability of the discrete cluster assignment step. Due to this, the gradients from the reconstruction loss cannot flow back into the encoder. A model with a similar problem is the recently proposed vector-quantized VAE (VQ-VAE) (van den Oord et al., 2017). It can be seen as being similar to a special case of our SOM-VAE model, where one sets $\beta = 0$, i.e. disables the SOM structure.

In order to mitigate the non-differentiability, the authors of the VQ-VAE propose to copy the gradients from $z_q$ to $z_e$. They acknowledge that this is an *ad hoc* approximation, but observed that it works well in their experiments. Due to our smaller number of embeddings compared to the VQ-VAE setup, the average distance between an encoding and its closest embedding is much larger in our case. The gradient copying (see above) thus ceases to be a feasible approximation, because the true gradients at points in the latent space which are farther apart will likely be very different.

In order to still overcome the non-differentiability issue, we propose to add the second reconstruction subterm to $\mathcal{L}_{\text{reconstruction}}$, where the reconstruction $\hat{x}_e$ is decoded directly from the encoding $z_e$. This adds a fully differentiable credit assignment path from the loss to the encoder and encourages $z_e$ to also be an informative representation of the input, which is a desirable model feature. Most importantly, it works well in practice (see Sec. 4.1).

Note that since $z_e$ is continuous and therefore much less constrained than $z_q$, this term is optimized easily and becomes small early in training. After that, mostly the $z_q$-term contributes to $\mathcal{L}_{\text{reconstruction}}$. One could therefore view the $z_e$-term as an initial encouragement to place the data encodings at sensible positions in the latent space, after which the actual clustering task dominates the training objective.

## 2.3 Encouraging Smoothness over Time

Our ultimate goal is to predict the development of time series in an interpretable way. This means that not only the state representations should be interpretable, but so should be the prediction as well. To this end, we use a temporal probabilistic model.

Learning a probabilistic model in a high-dimensional continuous space can be challenging. Thus, we exploit the low-dimensional discrete space induced by our SOM to learn a temporal model. For that, we define a system state as the assigned node in the SOM and then learn a Markov model for the transitions between those states. The model is learned jointly with the SOM-VAE, where the loss function becomes

$$\mathcal{L}(x^{t-1}, x^t, \hat{x}_q^t, \hat{x}_e^t) = \mathcal{L}_{\text{SOM-VAE}}(x^t, \hat{x}_q^t, \hat{x}_e^t) + \gamma \, \mathcal{L}_{\text{transitions}}(x^{t-1}, x^t) + \tau \, \mathcal{L}_{\text{smoothness}}(x^{t-1}, x^t) \qquad (2)$$

with weighting hyperparameters $\gamma$ and $\tau$.

The term $\mathcal{L}_{\text{transitions}}$ encourages the probabilities of actually observed transitions to be high. It is defined as $\mathcal{L}_{\text{transitions}}(x^{t-1}, x^t) = -\log P_M(z_q(x^{t-1}) \to z_q(x^t))$, with $P_M(z_q(x^{t-1}) \to z_q(x^t))$ being the probability of a transition from state $z_q(x^{t-1})$ to state $z_q(x^t)$ in the Markov model.

The term $\mathcal{L}_{\text{smoothness}}$ encourages the probabilities for transitions to nodes that are far away from the current data point to be low or respectively the nodes with high transition probabilities to be proximal. It achieves this by taking large values only for transitions to far away nodes that have a high probability under the model. It is defined as $\mathcal{L}_{\text{smoothness}}(x^{t-1}, x^t) = \mathbb{E}_{P_M(z_q(x^{t-1}) \to \tilde{e})} \left[ \|\tilde{e} - z_e(x^t)\|^2 \right]$. The probabilistic model can inform the evolution of the SOM through this term which encodes our prior belief that transitions in natural data happen smoothly and that future time points will therefore mostly be found in the neighborhood of previous ones. In a setting where the data measurements are noisy, this improves the clustering by acting as a temporal smoother.

## 3 Related Work

From the early inception of the *k-means* algorithm for clustering (Lloyd, 1982), there has been much methodological improvement on this unsupervised task. This includes methods that perform clustering in the latent

space of (variational) autoencoders (Aljalbout et al., 2018) or use a mixture of autoencoders for the clustering (Zhang et al., 2017; Locatello et al., 2018). The method most related to our work is the VQ-VAE (van den Oord et al., 2017), which can be seen as a special case of our framework (see above). Its authors have put a stronger focus on the discrete representation as a form of compression instead of clustering. Hence, our model and theirs differ in certain implementation considerations (see Sec. 2.2). All these methods have in common that they only yield a single number as a cluster assignment and provide no interpretable structure of relationships between clusters.

The self-organizing map (SOM) (Kohonen, 1990), however, is an algorithm that provides such an interpretable structure. It maps the data manifold to a lower-dimensional discrete space, which can be easily visualized in the 2D case. It has been extended to model dynamical systems (Barreto and Araujo, 2004) and combined with probabilistic models for time series (Sang et al., 2008), although without using learned representations. There are approaches to turn the SOM into a "deeper" model (Dittenbach et al., 2000), combine it with multi-layer perceptrons (Furukawa et al., 2005) or with metric learning (Płoński and Zaremba, 2012). However, it has (to the best of our knowledge) not been proposed to use SOMs in the latent space of (variational) autoencoders or any other form of unsupervised deep learning model.

Interpretable models for clustering and temporal predictions are especially crucial in fields where humans have to take responsibility for the model's predictions, such as in health care. The prediction of a patient's future state is an important problem, particularly on the *intensive care unit* (ICU) (Harutyunyan et al., 2017; Badawi et al., 2018). Probabilistic models, such as Gaussian processes, have been successfully applied in this domain (Colopy et al., 2016; Schulam and Arora, 2016). Recently, deep generative models have been proposed (Esteban et al., 2017), sometimes even in combination with probabilistic modeling (Lim and Schaar, 2018). To the best of our knowledge, SOMs have only been used to learn interpretable static representations of patients (Tirunagari et al., 2015), but not dynamic ones.

## 4 EXPERIMENTS

We performed experiments on MNIST handwritten digits (LeCun et al., 1998), Fashion-MNIST images of clothing (Xiao et al., 2017), synthetic time series of linear interpolations of those images, time series from a chaotic dynamical system and real world medical data from the *eICU Collaborative Research Database* (Goldberger et al., 2000). If not otherwise noted, we use the same architecture for all experiments, sometimes including the latent probabilistic model (*SOM-VAE_prob*) and sometimes excluding it (*SOM-VAE*). For model implementation details, we refer to the appendix (Sec. B)[1].

We found that our method achieves a superior clustering performance compared to other methods. We also show that we can learn a temporal probabilistic model concurrently with the clustering, which is on par with the maximum likelihood solution, while improving the clustering performance. Moreover, we can learn interpretable state representations of a chaotic dynamical system and discover patterns in real medical data.

### 4.1 CLUSTERING ON MNIST AND FASHION-MNIST

In order to test the clustering component of the SOM-VAE, we performed experiments on MNIST and Fashion-MNIST. We compare our model (including different adjustments to the loss function) against k-means (Lloyd, 1982) (`sklearn`-package (Pedregosa et al., 2011)), the VQ-VAE (van den Oord et al., 2017), a standard implementation of a SOM (`minisom`-package (Vettigli, 2017)) and our version of a GB-SOM (gradient-based SOM), which is a SOM-VAE where the encoder and decoder are set to be identity functions. The k-means algorithm was initialized using k-means++ (Arthur and Vassilvitskii, 2007). To ensure comparability of the performance measures, we used the same number of clusters (i.e. the same $k$) for all the methods.

The results of the experiment in terms of purity and normalized mutual information (NMI) are shown in Table 1. The SOM-VAE outperforms the other methods w.r.t. the clustering performance measures. It should be noted here that while k-means is a strong baseline, it is not density matching, i.e. the density of cluster centers is not proportional to the density of data points. Hence, the representation of data in a space induced by the k-means clusters can be misleading.

As argued in the appendix (Sec. C), NMI is a more balanced measure for clustering performance than purity. If one uses 512 embeddings in the SOM, one gets a lower NMI due to the penalty term for the number of

---

[1]Our code is available at `https://github.com/ratschlab/SOM-VAE`.

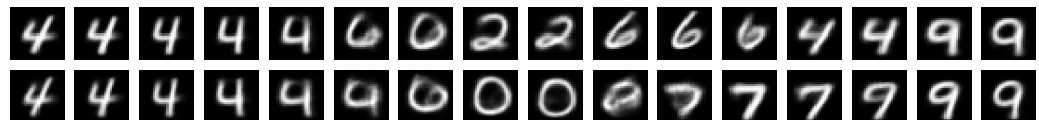

Figure 2: Images generated from a section of the SOM-VAE's latent space with 512 embeddings trained on MNIST. It yields a discrete two-dimensional representation of the data manifold in the higher-dimensional latent space.

Table 1: Performance comparison of our method and some baselines in terms of purity and normalized mutual information on different benchmark data sets. The methods marked with an asterisk are variants of our proposed method. The values are the means of 10 runs and the respective standard errors. Each method was used to fit 16 embeddings/clusters.

| Method | MNIST | | Fashion-MNIST | |
|---|---|---|---|---|
| | Purity | NMI | Purity | NMI |
| k-means | $0.690 \pm 0.000$ | $0.541 \pm 0.001$ | $0.654 \pm 0.001$ | $0.545 \pm 0.000$ |
| minisom | $0.406 \pm 0.006$ | $0.342 \pm 0.012$ | $0.413 \pm 0.006$ | $0.475 \pm 0.002$ |
| GB-SOM | $0.653 \pm 0.007$ | $0.519 \pm 0.005$ | $0.606 \pm 0.006$ | $0.514 \pm 0.004$ |
| VQ-VAE | $0.538 \pm 0.067$ | $0.409 \pm 0.065$ | $0.611 \pm 0.006$ | $0.517 \pm 0.002$ |
| no_grads* | $0.114 \pm 0.000$ | $0.001 \pm 0.000$ | $0.110 \pm 0.009$ | $0.018 \pm 0.016$ |
| gradcopy* | $0.583 \pm 0.004$ | $0.436 \pm 0.004$ | $0.556 \pm 0.008$ | $0.444 \pm 0.005$ |
| SOM-VAE* | $\mathbf{0.731 \pm 0.004}$ | $\mathbf{0.594 \pm 0.004}$ | $\mathbf{0.678 \pm 0.005}$ | $\mathbf{0.590 \pm 0.003}$ |

clusters, but it yields an interpretable two-dimensional representation of the manifolds of MNIST (Fig. 2, Supp. Fig. S4) and Fashion-MNIST (Supp. Fig. S5).

The experiment shows that the SOM in our architecture improves the clustering (SOM-VAE vs. VQ-VAE) and that the VAE does so as well (SOM-VAE vs. GB-SOM). Both parts of the model therefore seem to be beneficial for our task. It also becomes apparent that our reconstruction loss term on $z_e$ works better in practice than the gradient copying trick from the VQ-VAE (SOM-VAE vs. gradcopy), due to the reasons described in Section 2.2. If one removes the $z_e$ reconstruction loss and does not copy the gradients, the encoder network does not receive any gradient information any more and the learning fails completely (no_grads). Another interesting observation is that stochastically optimizing our SOM loss using Adam (Kingma and Ba, 2014) seems to discover a more performant solution than the classical SOM algorithm (GB-SOM vs. minisom). This could be due to the dependency of the step size on the distance between embeddings and encodings, as described in Section 2.1. Since k-means seems to be the strongest competitor, we are including it as a reference baseline in the following experiments as well.

## 4.2 MARKOV TRANSITION MODEL ON THE DISCRETE REPRESENTATIONS

In order to test the probabilistic model in our architecture and its effect on the clustering, we generated synthetic time series data sets of (Fashion-)MNIST images being linearly interpolated into each other. Each time series consists of 64 frames, starting with one image from (Fashion-)MNIST and smoothly changing sequentially into four other images over the length of the time course.

After training the model on these data, we constructed the maximum likelihood estimate (MLE) for the Markov model's transition matrix by fixing all the weights in the SOM-VAE and making another pass over the training set, counting all the observed transitions. This MLE transition matrix reaches a negative log likelihood of $0.24$, while our transition matrix, which is learned concurrently with the architecture, yields $0.25$. Our model is therefore on par with the MLE solution.

Comparing these results with the clustering performance on the standard MNIST and Fashion-MNIST test sets, we observe that the performance in terms of NMI is not impaired by the inclusion of the probabilistic model into the architecture (Tab. 2). On the contrary, the probabilistic model even slightly increases the performance on Fashion-MNIST. Note that we are using 64 embeddings in this experiment instead of 16, leading to a higher clustering performance in terms of purity, but a slightly lower performance in terms of NMI compared to Table 1. This shows again that the measure of purity has to be interpreted with care when comparing

Table 2: Performance comparison of the SOM-VAE with and without latent Markov model (SOM-VAE-prob) against k-means in terms of purity and normalized mutual information on different benchmark data sets. The values are the means of 10 runs and the respective standard errors. Each method is used to fit 64 embeddings/clusters.

| Method | MNIST | | Fashion-MNIST | |
|---|---|---|---|---|
| | Purity | NMI | Purity | NMI |
| k-means | $0.791 \pm 0.005$ | $0.537 \pm 0.001$ | $0.703 \pm 0.002$ | $0.492 \pm 0.001$ |
| SOM-VAE | $\mathbf{0.868 \pm 0.003}$ | $\mathbf{0.595 \pm 0.002}$ | $\mathbf{0.739 \pm 0.002}$ | $0.520 \pm 0.002$ |
| SOM-VAE-prob | $0.858 \pm 0.004$ | $\mathbf{0.596 \pm 0.001}$ | $0.724 \pm 0.003$ | $\mathbf{0.525 \pm 0.002}$ |

different experimental setups and that therefore the normalized mutual information should be preferred to make quantitative arguments.

This experiment shows that we can indeed fit a valid probabilistic transition model concurrently with the SOM-VAE training, while at the same time not hurting the clustering performance. It also shows that for certain types of data the clustering performance can even be improved by the probabilistic model (see Sec. 2.3).

### 4.3 INTERPRETABLE REPRESENTATIONS OF CHAOTIC TIME SERIES

In order to assess whether our model can learn an interpretable representation of more realistic chaotic time series, we train it on synthetic trajectories simulated from the famous *Lorenz system* (Lorenz, 1963). The Lorenz system is a good example for this assessment, since it offers two well defined macro-states (given by the attractor basins) which are occluded by some chaotic noise in the form of periodic fluctuations around the attractors. A good interpretable representation should therefore learn to largely ignore the noise and model the changes between attractor basins. For a review of the Lorenz system and details about the simulations and the performance measure, we refer to the appendix (Sec. D.2).

In order to compare the interpretability of the learned representations, we computed entropy distributions over simulated subtrajectories in the real system space, the attractor assignment space and the representation spaces for k-means and our model. The computed entropy distributions over all subtrajectories in the test set are depicted in Figure 3.

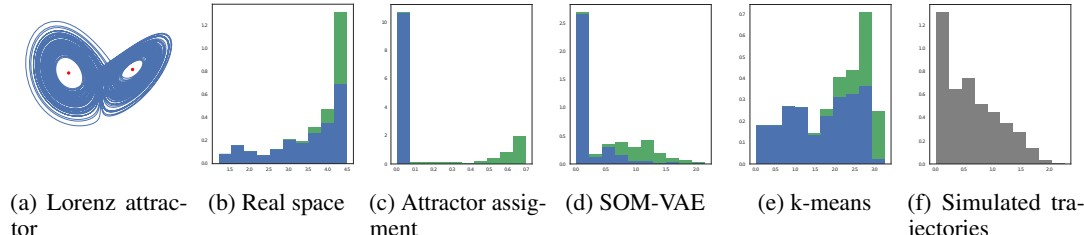

(a) Lorenz attractor    (b) Real space    (c) Attractor assignment    (d) SOM-VAE    (e) k-means    (f) Simulated trajectories

Figure 3: Histograms of entropy distributions (entropy on the x-axes) over all Lorenz attractor subtrajectories [a] of 100 time steps length in our test set. Subtrajectories without a change in attractor basin are colored in blue, the ones where a change has taken place in green.

The experiment shows that the SOM-VAE representations (Fig. 3d) are much closer in entropy to the ground-truth attractor basin assignments (Fig. 3c) than the k-means representations (Fig. 3e). For most of the subtrajectories without attractor basin change they assign a very low entropy, effectively ignoring the noise, while the k-means representations partially assign very high entropies to those trajectories. In total, the k-means representations' entropy distribution is similar to the entropy distribution in the noisy system space (Fig. 3b). The representations learned by the SOM-VAE are therefore more interpretable than the k-means representations with regard to this interpretability measure. As could be expected from these figures, the SOM-VAE representation is also superior to the k-means one in terms of purity with respect to the attractor assignment (0.979 vs. 0.956) as well as NMI (0.577 vs. 0.249).

Finally, we use the learned probabilistic model on our SOM-VAE representations to sample new latent system trajectories and compute their entropies. The distribution looks qualitatively similar to the one over real

Table 3: Performance comparison of our method with and without probabilistic model (SOM-VAE-prob and SOM-VAE) against k-means in terms of normalized mutual information on a challenging unsupervised prediction task on real eICU data. The dynamic endpoints are the maximum of the physiology score within the next 6, 12 or 24 hours (*physiology_6_hours*, *physiology_12_hours*, *physiology_24_hours*). The values are the means of 10 runs and the respective standard errors. Each method is used to fit 64 embeddings/clusters.

| Method | physiology_6_hours | physiology_12_hours | physiology_24_hours |
|---|---|---|---|
| k-means | $0.0411 \pm 0.0007$ | $0.0384 \pm 0.0006$ | $0.0366 \pm 0.0005$ |
| SOM-VAE | $0.0407 \pm 0.0005$ | $0.0376 \pm 0.0004$ | $0.0354 \pm 0.0004$ |
| SOM-VAE-prob | $\mathbf{0.0474 \pm 0.0006}$ | $\mathbf{0.0444 \pm 0.0006}$ | $\mathbf{0.0421 \pm 0.0005}$ |

(a) k-means     (b) VQ-VAE     (c) SOM-VAE-prob     (d) Patient trajectories

Figure 4: Comparison of the patient state representations learned by different models. The clusters are colored by degree of patient abnormality as measured by a variant of the APACHE physiology score (more yellow means "less healthy"). White squares correspond to unused clusters, i.e. clusters that contain less than 0.1 percent of the data points. Subfigure (d) shows two patient trajectories in the SOM-VAE-prob representation over their respective whole stays in the ICU. The dots mark the ICU admission, the stars the discharge from the ICU (cured [green] or dead [red]). It can be seen that our model is the only one that learns a topologically interpretable structure.

trajectories (Fig. 3), but our model slightly overestimates the attractor basin change probabilities, leading to a heavier tail of the distribution.

## 4.4 LEARNING REPRESENTATIONS OF REAL MEDICAL TIME SERIES

In order to demonstrate interpretable representation learning on a complex real world task, we trained our model on vital sign time series measurements of intensive care unit (ICU) patients. We analyze the performance of the resulting clustering w.r.t. the patients' future physiology states in Table 3. This can be seen as a way to assess the representations' informativeness for a downstream prediction task. For details regarding the data selection and processing, we refer to the appendix (Sec. D.3).

Our full model (including the latent Markov model) performs best on the given tasks, i.e. better than k-means and also better than the SOM-VAE without probabilistic model. This could be due to the noisiness of the medical data and the probabilistic model's smoothing tendency (see Sec. 2.3).

In order to qualitatively assess the interpretability of the probabilistic SOM-VAE, we analyzed the average future physiology score per cluster (Fig. 4). Our model exhibits clusters where higher scores are enriched compared to the background level. Moreover, these clusters form compact structures, facilitating interpretability. We do not observe such interpretable structures in the other methods. For full results on acute physiology scores, an analogue experiment showing the future mortality risk associated with different regions of the map, and an analysis of enrichment for particular physiological abnormalities, we refer to the appendix (Sec. D.4).

As an illustrative example for data visualization using our method, we show the trajectories of two patients that start in the same state (Fig. 4d). The trajectories are plotted in the representation space of the probabilistic SOM-VAE and should thus be compared to the visualization in Figure 4c. One patient (*green*) stays in the regions of the map with low average physiology score and eventually gets discharged from the hospital healthily. The other one (*red*) moves into map regions with high average physiology score and ultimately dies. Such knowledge could be helpful for doctors, who could determine the risk of a patient for certain deterioration scenarios from a glance at their trajectory in the SOM-VAE representation.

## 5    CONCLUSION

The SOM-VAE can recover topologically interpretable state representations on time series and static data. It provides an improvement to standard methods in terms of clustering performance and offers a way to learn discrete two-dimensional representations of the data manifold in concurrence with the reconstruction task. It introduces a new way of overcoming the non-differentiability of the discrete representation assignment and contains a gradient-based variant of the traditional self-organizing map that is more performant than the original one. On a challenging real world medical data set, our model learns more informative representations with respect to medically relevant prediction targets than competitor methods. The learned representations can be visualized in an interpretable way and could be helpful for clinicians to understand patients' health states and trajectories more intuitively.

It will be interesting to see in future work whether the probabilistic component can be extended to not just improve the clustering and interpretability of the whole model, but also enable us to make predictions. Promising avenues in that direction could be to increase the complexity by applying a higher order Markov Model, a Hidden Markov Model or a Gaussian Process. Another fruitful avenue of research could be to find more theoretically principled ways to overcome the non-differentiability and compare them with the empirically motivated ones. Lastly, one could explore deviating from the original SOM idea of fixing a latent space structure, such as a 2D grid, and learn the neighborhood structure as a graph directly from data.

### ACKNOWLEDGMENTS

FL is supported by the Max Planck/ETH Center for Learning Systems. MH is supported by the Grant No. 205321_176005 "Novel Machine Learning Approaches for Data from the Intensive Care Unit" of the Swiss National Science Foundation (to GR). VF, FL, MH and HS are partially supported by ETH core funding (to GR). We thank Natalia Marciniak for her administrative efforts; Marc Zimmermann for technical support; Gideon Dresdner, Stephanie Hyland, Viktor Gal, Maja Rudolph and Claire Vernade for helpful discussions; and Ron Swanson for his inspirational attitude.

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

APPENDIX

## A   Self-organizing maps

The general idea of a self-organizing map (SOM) is to approximate a data manifold in a high-dimensional continuous space with a lower dimensional discrete one (Kohonen, 1990). It can therefore be seen as a nonlinear discrete dimensionality reduction. The mapping is achieved by a procedure in which this discrete representation (the *map*) is randomly embedded into the data space and then iteratively optimized to approach the data manifold more closely.

The map consists of $k$ nodes $V = \{v_1, \ldots, v_k\}$, where every node corresponds to an embedding in the data space $e_v \in \mathbb{R}^d$ and a representation in the lower-dimensional discrete space $m_v \in M$, where usually $M \subset \mathbb{N}^2$. There are two different geometrical measures that have to be considered during training: the neighborhood function $N(m_u, m_{\tilde{v}})$ that is defined on the low-dimensional map space and the Euclidean distance $D(e_u, e_{\tilde{v}}) = \|e_u - e_{\tilde{v}}\|_2$ in the high-dimensional data space. The SOM optimization tries to induce a coupling between these two properties, such that the topological structure of the representation reflects the geometrical structure of the data.

---

**Algorithm 1** Self-organizing map training

---

**Require:** data set $\mathcal{D} = \{x_1, \ldots, x_n \mid x_i \in \mathbb{R}^d\}$, number of nodes $k$, neighborhood function $N(\cdot)$, step size $\eta$
    initialize set of $k$ nodes $V = \{v_1, \ldots, v_k\}$
    initialize embeddings $e_v \in \mathbb{R}^d \ \forall \ v \in V$
    **while** not converged **do**
        **for all** $x_i \in \mathcal{D}$ **do**
            find the closest SOM node $\tilde{v} := \arg\min_{v \in V} \|x_i - e_v\|_2$
            update node embedding $e_{\tilde{v}} \leftarrow e_{\tilde{v}} + \eta\,(x_i - e_{\tilde{v}})$
            **for all** $u \in V \backslash \tilde{v}$ **do**
                update neighbor embedding $e_u \leftarrow e_u + \eta\,N(m_{\tilde{v}}, m_u)(x_i - e_u)$
            **end for**
        **end for**
    **end while**

---

The SOM training procedure is described in Algorithm 1. During training on a data set $\mathcal{D}$, a winner node $\tilde{v}$ is chosen for every point $x_i$ according to the Euclidean distance of the point and the node's embedding in the data space. The embedding vector for the winner node is then updated by pulling it into the direction of the data point with some step size $\eta$. The embedding vectors of the other nodes are also updated – potentially with a smaller step size – depending on whether they are neighbors of the winner node in the map space $M$.

The neighborhood is defined by the neighborhood function $N(m_u, m_{\tilde{v}})$. There can be different design choices for the neighborhood function, e.g. rectangular grids, hexagonal grids or Gaussian neighborhoods. For simplicity and ease of visualization, we usually choose a two-dimensional rectangular grid neighborhood in this paper.

In this original formulation of the SOM training, the nodes are updated one by one with a fixed step size. In our model, however, we use a gradient-based optimization of their distances to the data points and update them in minibatches. This leads to larger step sizes when they are farther away from the data and smaller step sizes when they are close. Overall, our gradient-based SOM training seems to perform better than the original formulation (see Tab. 1).

It also becomes evident from this procedure that it will be very hard for the map to fit disjoint manifolds in the data space. Since the nodes of the SOM form a fully connected graph, they do not possess the ability to model spatial gaps in the data. We overcome this problem in our work by mapping the data manifold with a variational autoencoder into a lower-dimensional latent space. The VAE can then learn to close the aforementioned gaps and map the data onto a compact latent manifold, which can be more easily modeled with the SOM.

## B  IMPLEMENTATION DETAILS

The hyperparameters of our model were optimized using Robust Bayesian Optimization with the packages `sacred` and `labwatch` (Greff et al., 2017) for the parameter handling and `RoBo` (Klein et al., 2017) for the optimization, using the mean squared reconstruction error as the optimization criterion. Especially the weighting hyperparameters $\alpha, \beta, \gamma$ and $\tau$ (see Eq. (1) and Eq. (2)) have to be tuned carefully, such that the different parts of the model converge at roughly the same rate. We found that 2000 steps of Bayesian optimization sufficed to yield a performant hyperparameter assignment.

Since our model defines a general framework, some competitor models can be seen as special cases of our model, where certain parts of the loss function are set to zero or parts of the architecture are omitted. We used the same hyperparameters for those models. For external competitor methods, we used the hyperparameters from the respective publications where applicable and otherwise the default parameters from their packages. The models were implemented in TensorFlow (Abadi et al., 2016) and optimized using Adam (Kingma and Ba, 2014).

## C  CLUSTERING PERFORMANCE MEASURES

Given that one of our most interesting tasks at hand is the clustering of data, we need some performance measures to objectively compare the quality of this clustering with other methods. The measures that we decided to use and that have been used extensively in the literature are *purity* and *normalized mutual information* (NMI) (Manning et al., 2008). We briefly review them in the following.

Let the set of ground truth classes in the data be $C = \{c_1, c_2, \ldots, c_J\}$ and the set of clusters that result from the algorithm $\Omega = \{\omega_1, \omega_2, \ldots, \omega_K\}$. The *purity* $\pi$ is then defined as $\pi(C, \Omega) = \frac{1}{N} \sum_{k=1}^{K} \max_j |\omega_k \cap c_j|$ where $N$ is the total number of data points. Intuitively, the purity is the accuracy of the classifier that assigns the most prominent class label in each cluster to all of its respective data points.

While the purity has a very simple interpretation, it also has some shortcomings. One can for instance easily observe that a clustering with $K = N$, i.e. one cluster for every single data point, will yield a purity of $1.0$ but still probably not be very informative for most tasks. It would therefore be more sensible to have another measure that penalizes the number of clusters. The normalized mutual information is one such measure.

The NMI is defined as $NMI(C, \Omega) = \frac{2\,I(C, \Omega)}{H(C) + H(\Omega)}$ where $I(C, \Omega)$ is the mutual information between $C$ and $\Omega$ and $H(\cdot)$ is the Shannon information entropy. While the entropy of the classes is a data-dependent constant, the entropy of the clustering increases with the number of clusters. It can therefore be seen as a penalty term to regularize the trade-off between low intra-cluster variance and a small number of clusters. Both NMI and purity are normalized, i.e. take values in $[0, 1]$.

## D  EXPERIMENTAL DETAILS

### D.1  CLUSTERING ON MNIST AND FASHION-MNIST

Additionally to the results in Table 1, we performed experiments to assess the influence of the number of clusters $k$ on the clustering performance of our method. We chose different values for $k$ between 4 and 64 and tested the clustering performance on MNIST and Fashion-MNIST (Tab. S1).

It can be seen that the purity increases monotonically with $k$, since it does not penalize larger numbers of clusters (see Sec. C). The NMI, however, includes an automatic penalty for misspecifying the model with too many clusters. It therefore increases first, but then decreases again for too large values of $k$. The optimal $k$ according to the NMI seems to lie between 16 and 36.

### D.2  INTERPRETABLE REPRESENTATIONS OF CHAOTIC TIME SERIES

The Lorenz system is the system of coupled ordinary differential equations defined by

$$\frac{dX}{dt} = a(Y - X) \qquad \frac{dY}{dt} = X(b - Z) - Y \qquad \frac{dZ}{dt} = XY - cZ$$

Table S1: Performance comparison of our method with different numbers of clusters in terms of purity and normalized mutual information on different benchmark data sets. The values are the means of 10 runs and the respective standard errors.

| Number of clusters | MNIST | | Fashion-MNIST | |
|---|---|---|---|---|
| | Purity | NMI | Purity | NMI |
| $k = 4$ | $0.364 \pm 0.009$ | $0.378 \pm 0.018$ | $0.359 \pm 0.005$ | $0.431 \pm 0.008$ |
| $k = 9$ | $0.626 \pm 0.006$ | $0.554 \pm 0.004$ | $0.558 \pm 0.007$ | $0.560 \pm 0.006$ |
| $k = 16$ | $0.721 \pm 0.006$ | $0.587 \pm 0.003$ | $0.684 \pm 0.003$ | $\mathbf{0.589 \pm 0.003}$ |
| $k = 25$ | $0.803 \pm 0.003$ | $\mathbf{0.613 \pm 0.002}$ | $0.710 \pm 0.003$ | $0.572 \pm 0.002$ |
| $k = 36$ | $0.850 \pm 0.002$ | $\mathbf{0.612 \pm 0.001}$ | $0.732 \pm 0.002$ | $0.556 \pm 0.002$ |
| $k = 49$ | $0.875 \pm 0.002$ | $0.608 \pm 0.001$ | $0.750 \pm 0.002$ | $0.545 \pm 0.001$ |
| $k = 64$ | $\mathbf{0.894 \pm 0.002}$ | $0.599 \pm 0.001$ | $\mathbf{0.758 \pm 0.002}$ | $0.532 \pm 0.001$ |

with tuning parameters $a$, $b$ and $c$. For parameter choices $a = 10$, $b = 28$ and $c = \frac{8}{3}$, the system shows chaotic behavior by forming a strange attractor (Tucker, 1999) with the two attractor points being given by $p_{1,2} = [\pm\sqrt{c(b-1)}, \pm\sqrt{c(b-1)}, b-1]^T$.

We simulated 100 trajectories of 10,000 time steps each from the chaotic system and trained the SOM-VAE as well as k-means on it with 64 clusters/embeddings respectively. The system chaotically switches back and forth between the two attractor basins. By computing the Euclidian distance between the current system state and each of the attractor points $p_{1,2}$, we can identify the current attractor basin at each time point.

In order to assess the interpretability of the learned representations, we have to define an objective measure of interpretability. We define interpretability as the similarity between the representation and the system's ground truth macro-state. Since representations at single time points are meaningless with respect to this measure, we compare the evolution of representations and system state over time in terms of their entropy.

We divided the simulated trajectories from our test set into spans of 100 time steps each. For every subtrajectory, we computed the entropies of those subtrajectories in the real system space (macro-state and noise), the assigned attractor basin space (noise-free ground-truth macro-state), the SOM-VAE representation and the k-means representation. We also observed for every subtrajectory whether or not a change between attractor basins has taken place. Note that the attractor assignments and representations are discrete, while the real system space is continuous. In order to make the entropies comparable, we discretize the system space into unit hypercubes for the entropy computation. For a representation $\mathcal{R}$ with assignments $\mathcal{R}_t$ at time $t$ and starting time $t_{start}$ of the subtrajectory, the entropies are defined as

$$H(\mathcal{R}, t_{start}) = H(\{R_t \mid t_{start} \leq t < t_{start} + 100\}) \tag{3}$$

with $H(\cdot)$ being the Shannon information entropy of a discrete set.

### D.3 LEARNING REPRESENTATIONS OF ACUTE PHYSIOLOGICAL STATES IN THE ICU

All experiments were performed on dynamic data extracted from the `eICU` Collaborative Research Database (Goldberger et al., 2000). Irregularly sampled time series data were extracted from the raw tables and then resampled to a regular time grid using a combination of forward filling and missing value imputation using global population statistics. We chose a grid interval of one hour to capture the rapid dynamics of patients in the ICU.

Each sample in the time-grid was then labeled using a dynamic variant of the `APACHE` score (Knaus et al., 1985), which is a proxy for the instantaneous physiological state of a patient in the ICU. Specifically, the variables `MAP`, `Temperature`, `Respiratory rate`, `HCO3`, `Sodium`, `Potassium`, and `Creatinine` were selected from the score definition, because they could be easily defined for each sample in the `eICU` time series. The value range of each variable was binned into ranges of normal and abnormal values, in line with the definition of the `APACHE` score, where a higher score for a variable is obtained for abnormally high or low values. The scores were then summed up, and we define the predictive score as the *worst (highest) score* in the next $t$ hours, for $t \in \{6, 12, 24\}$. Patients are thus stratified by their expected pathology in the near future, which corresponds closely to how a physician would perceive the state of a patient. The training set consisted of 7000 unique patient stays, while the test set contained 3600 unique stays.

### D.4 DETAILED ANALYSIS OF SOMVAEPROB PATIENT STATES

As mentioned in the main text (see Fig 4c) the `SOMVAEProb` is able to uncover compact and interpretable structures in the latent space with respect to future physiology scores. In this section we show results for acute physiology scores in greater detail, analyze enrichment for future mortality risk, arguably the most important severity indicator in the ICU, and explore phenotypes for particular physiological abnormalities.

FULL RESULTS FOR FUTURE ACUTE PHYSIOLOGY SCORES

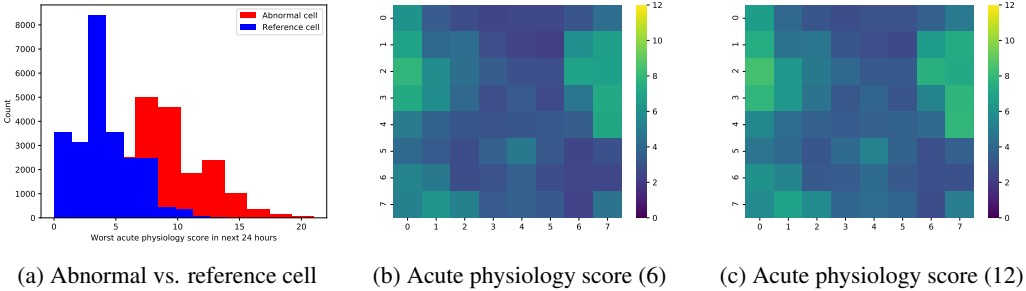

(a) Abnormal vs. reference cell    (b) Acute physiology score (6)    (c) Acute physiology score (12)

Figure S1: (a) shows the difference in distribution of the acute physiology score in the next 24 hours, between time-points assigned to the most abnormal cell in the `SOMVAEprob` map with coordinates [2,0] vs. a normal cell chosen from the middle of the map with coordinates [4,3]. It is apparent that the distributions are largely disjoint, which means that the representation induced by `SOMVAEprob` clearly distinguishes these risk profiles. Statistical tests for difference in distribution and location parameter are highly significant at p-values of $p \leq 10^{-3}$, as we have validated using a 2-sample $t$-test and Kolmogorov-Smirnov test. In (b-c) the enrichment of the map for the mean acute physiology score in the next 6 and 12 hours is shown, for completeness. The enrichment patterns on the 3 maps, for the future horizons $\{6, 12, 24\}$, are almost identical, which provides empirical evidence for the temporal stability of the `SOMVAEProb` embedding.

DYNAMIC MORTALITY RISK OF PATIENTS ON THE SOMVAEPROB MAP

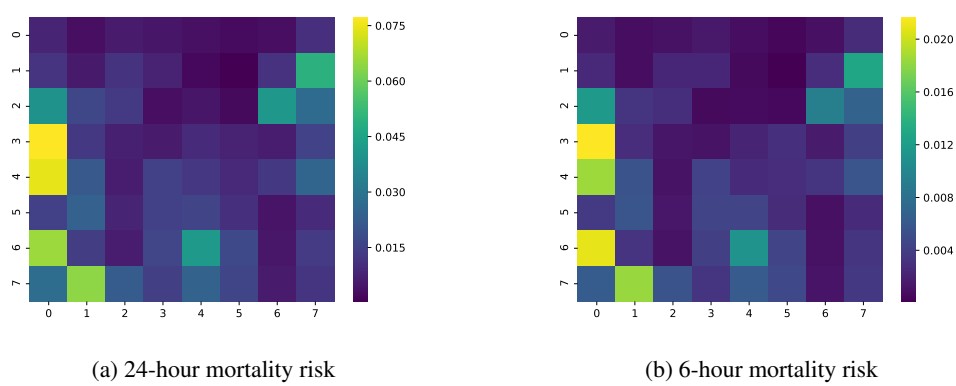

(a) 24-hour mortality risk    (b) 6-hour mortality risk

Figure S2: (a) Dynamic mortality risk in the next 24 hours. (b) Short-term dynamic mortality risk in the next 6 hours. We observe that the left-edge and right-edge regions of the `SOMVAEprob` map which are enriched for higher acute physiology scores (see Fig 4c) also exhibit elevated mortality rates over the baseline. Interestingly, according to future mortality risk, which is an important severity indicator, patients on the left-edge are significantly more sick on average than those on the right edge, which is less visible from the enrichment for acute physiology scores.

PATIENT STATE PHENOTYPES ON THE SOMVAEPROB MAP

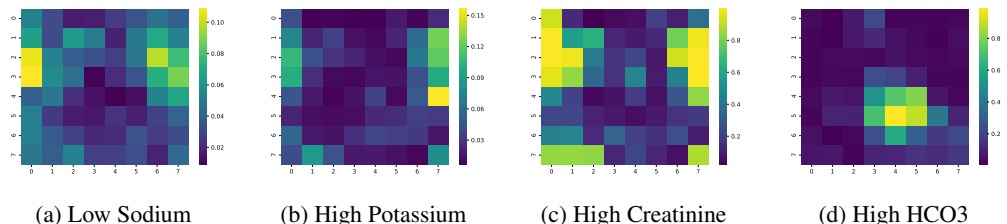

    (a) Low Sodium        (b) High Potassium       (c) High Creatinine      (d) High HCO3

Figure S3: (a) Prevalence of abnormally low sodium lab value in the next 24 hours, (b-d) Prevalence of abnormally high potassium/creatinine/HCO3 lab values in the next 24 hours. Each sub-figure illustrates the enrichment of a distinct phenotype on the SOMVAEprob map. Low sodium and high potassium states are enriched near the left edge, and near the right edge, respectively, which could represent sub-types of the high-risk phenotype found in these regions (compare Fig 4c for the distribution of the acute physiology score). Elevated creatinine is a trait that occurs in both these regions. A compact structure associated with elevated HCO3 can be found in the center of the map, which could represent a distinct phenotype with lower mortality risk in our cohort. In all phenotypes, the tendency of SOMVAEprob to recover compact structures is exemplified.

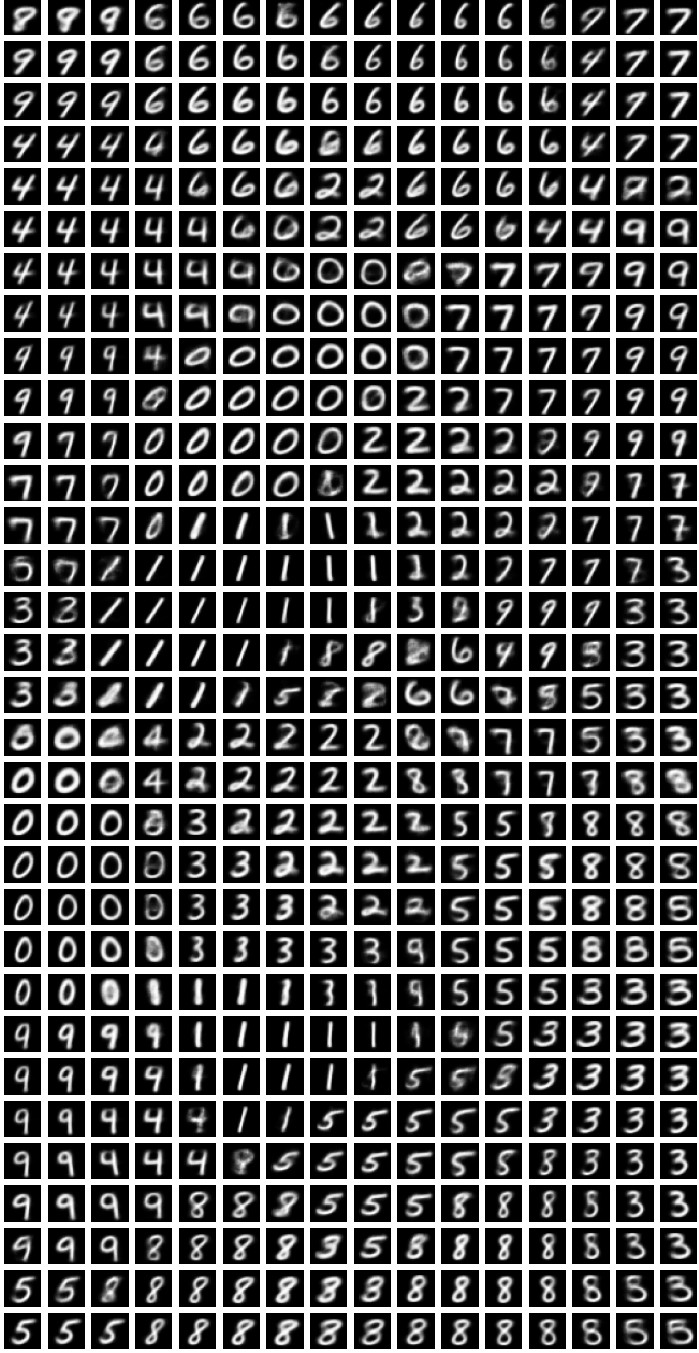

Figure S4: Images generated from the SOM-VAE's latent space with 512 embeddings trained on MNIST. It yields an interpretable discrete two-dimensional representation of the data manifold in the higher-dimensional latent space.

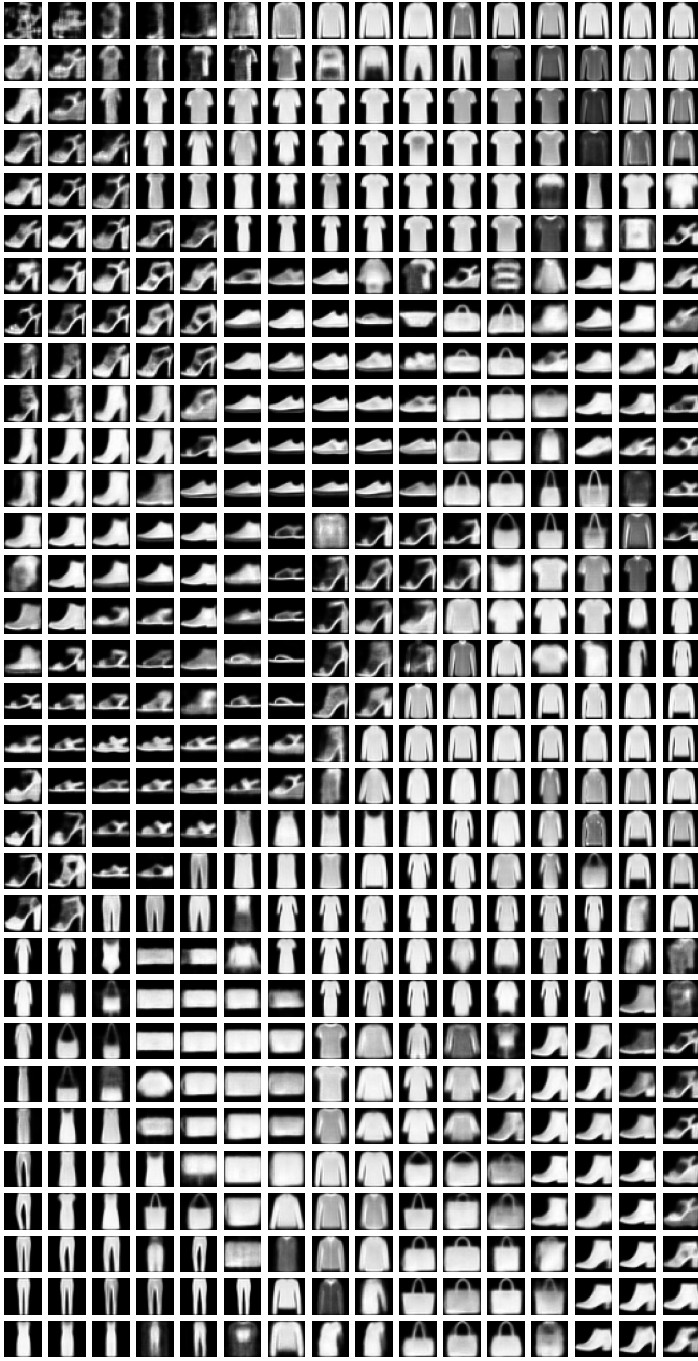

Figure S5: Images generated from the SOM-VAE's latent space with 512 embeddings trained on Fashion-MNIST. It yields an interpretable discrete two-dimensional representation of the data manifold in the higher-dimensional latent space.

