# OpenReview forum: "SOM-VAE: Interpretable Discrete Representation Learning on Time Series"
_ICLR.cc/2019/Conference_

### Official Review · AnonReviewer2 · 2018-10-26
**Interesting work, methodological aspects and implementation details to be clarified. Improving comparison with state of the art**

**Rating:** 6
**Confidence:** 4

**Review:**

This work addresses the problem of learning latent embeddings of high-dimensional time series data. The paper emphasises the need of interpretable representations accounting for the correlated nature of temporal data. To this scope, the study proposes to cluster the data in a latent space estimated through an auto-encoder. The clustering is obtained by leveraging on the idea of self-organising maps (SOM). Within this setting, the data is mapped into a 2D lattice where each coordinate point represents the center of an inner cluster.
This construction motivates the formulation of the auto encoder through the definition of several cost terms promoting reconstruction, clustering, and consistency across latent mappings.
This definition of the problem allows an heuristic for circumventing the non-differentiability of the discrete mapping. The enhance consistency over time, the model is further equipped with an additional cost term enforcing transition smoothness across data points and latent embeddings.

The experiments are carried out with respect to synthetic 2D time-series, chaotic time-series from dynamical systems, and clinical data. In each case the proposed method shows promising results with respect to the proposed benchmark.

The study presents some interesting methodological and technical ideas. On the other hand the manuscript presentation is quite convoluted, at the expense of a lacks of clarity in the details about the implementation of the methodology. Moreover, motivated by practical aspects, the model optimisation relies on computational strategies not completely supported from the theoretical point of view (such as the zeroing of the gradient in backpropagation, or the approximation of the clustering function to overcome non-differentiability). The impact of these modeling choices would deserve more investigation and discussion.

Detailed comments:

- As also stated by the authors, the use of a 2D latent representation is completely arbitrary. It may be true that a 2D embedding provides a simple visualisation, however interpretability can be obtained also with much richer representations in a number of different ways (e.g. sparsity, parametric representations, …). Therefore the feeling is that the proposed structure may be quite ad-hoc, and one may wonder whether the algorithm would still generalise to more complex latent representations.
- Related to the previous comment, the number of latent points seems to be crucial to the performance of the method. However this aspect is not discussed in detail, while it would be beneficial to provide experiment about the sensitivity and accuracy with respect to the choice if this parameters.
- The method relies on several cost terms plugged together. While each of them takes care of specific consistency aspects of the model, their mutual relation and balance may be very critical. This is governed by a series of trade-off parameters whose effect is not discussed  nor explored throughout the study. I guess that the optimisation stability may be also quite sensitive to this trade-off, and it would be important to provide more details about this aspect.
- Surprisingly, k-means seems to perform quite well in spite of its simplicity. Also, there is no mention about initialisation and choice of the parameter “k”. The authors may want to better discuss the performance of this algorithm, especially compared to its much lower modeling complexity with respect to the proposed method.
- Still related to the comparison with respect to the state-of-art, interpretability in time series analysis can be achieved with much lesser assumptions and parameters by using standard approaches such as independent component analysis. I would expect this sort of comparison, especially in case of long-term data such as the one provided in the Lorenz system.
- Clustering of short-term time series, such as the clinical ones, is a challenging task. The feeling is that a highly parametrised model, such as the proposed one,  may still not be superior with respect to classical methods, such as the mixture of linear regressions. This sort of comparison would be quite informative to appreciate the real value of the proposed methodology.

---

### Official Review · AnonReviewer3 · 2018-11-05
**A representation learning method for time series data**

**Rating:** 6
**Confidence:** 2

**Review:**

This paper proposes a deep learning method for representation learning in time series data. The goal is to learn a discrete two-dimensional representation of the time series data in an interpretable manner. The model is constructed on the basis of self-organizing maps (SOM) and involves reconstruction error in the training. In order to address the non-differentiability in the discrete representation assignment, the authors propose to include an extra reconstruction loss term w.r.t. the discrete representation. The authors conduct experiments on both static and time series data and validate that the method perform better than related methods in terms of clustering results as well as interpretability.

This paper deals with an interesting problem as learning an interpretable representation in time series data is important in areas such as health care and business. However, I am afraid the presentation of this paper is a bit difficult to follow. Some concerns/questions as below:

1) As the paper is based on SOM, some illustration of this method would be helpful for readers to understand the idea and learn the major contribution;

2) The authors use NMI and purity to evaluate the clustering performance. I was curious why not use the clustering accuracy as well?

3) Some more explanation on Fig. 4(d) would be helpful.

---

### Official Review · AnonReviewer1 · 2018-11-05
**Very nice research**

**Rating:** 9
**Confidence:** 4

**Review:**

This paper proposes a novel clustering technique that combines the self-organising map (SOM) (Kohonen, 1998) ideas with the differentiable quantized clustering ideas of VQ-VAE (van den Oord et al, 2017). The resulting algorithm is able to achieve better unsupervised clustering than either technique on its own. It also beats the k-means clustering approach. The authors also suggest augmenting their setup with a model of cluster transition dynamics for time-series data, which seems to improve the clustering further, as well as providing an interpretable 2D visualisation of the system's dynamics.

This approach addresses an important problem of easy interpretable visualisation of complex dynamics of a multi-dimensional system. This solution can have immediate wide spread real life applications, for example in fields like medicine or finance. The paper is very well written and the model clearly outperforms its baselines. The authors also include very nice evaluation of the importance of the different parts of the model for the final performance.

This is one of the best papers I have reviewed in a while. The only question I have is in terms of the medical data. The map learnt by SOM-VAE-prob presented in Fig. 4 appears to have 2 clusters with 'less healthy' patients (near the top left and top right edges). It would be good to have an analysis of what differences there are between these two clusters, and whether they are recovered consistently.

---

### Public Comment · ~Chun-Hao_Chang1 · 2019-05-16
**All the rebutal deleted or there is no rebutal at all?**

I am wondering why I don't see any rebuttal in this paper. From the meta-review, it seems that those comment did exist. So is it deleted? It very significantly reduces the comprehension of the paper.

---

> ### Comment · Area_Chair1 · 2019-05-16
> **Rebuttal not public**
>
> The authors chose to make their rebuttals privately to the reviewers, which is something that OpenReview supports.

---

### Meta-Review · Area_Chair1 · 2018-12-17
**Solid contributions, strong results, well-written**

**Confidence:** 4
**Recommendation:** Accept (Poster)

**Metareview:**

This paper combines probabilistic models, VAEs, and self-organizing maps to learn interpretable representations on time series. The proposed contributions are a novel and interesting combination of existing ideas, in particular, the extension to time-series data by modeling the cluster dynamics. The empirical results show improved unsupervised clustering performance, on both synthetic and real datasets, compared to a number of baselines. The resulting 2D embedding also provides an interpretable visualization.

The reviewers and the AC identified a number of potential weaknesses in the presentation in the original submission: (1) there was insufficient background on SOMs, leaving the readers unable to comprehend the contributions, (2) some of the details about the experiments were missing, such as how the baselines were constructed, (3) additional experiments were needed in regards to the hyper-parameters, such as number of clusters and the weighting in the loss, and (4) Figure 4d required a description of the results.

The revision and the comments by the authors addressed most of these comments, and the reviewers felt that their concerns had been alleviated.

Thus, the reviewers felt the paper should be accepted.